# Flexible Free-Standing Cu*_x_*O/Ag_2_O (*x* = 1, 2) Nanowires Integrated with Nanoporous Cu-Ag Network Composite for Glucose Sensing

**DOI:** 10.3390/nano10020357

**Published:** 2020-02-19

**Authors:** Qian Zhang, Man Li, Chunling Qin, Zhifeng Wang, Weimin Zhao, Yongyan Li

**Affiliations:** School of Materials Science and Engineering, Hebei University of Technology, Tianjin 300130, China; qianzhanghebut@163.com (Q.Z.); mlimail2017@163.com (M.L.); zfwangmail@163.com (Z.W.);

**Keywords:** bimetallic, nanoporous, Cu*_x_*O/Ag_2_O nanowires, flexibility, glucose sensor

## Abstract

To improve glucose electrocatalytic performance, one efficient manner is to develop a novel Cu-Ag bimetallic composite with fertile porosity and unique architecture. Herein, the self-supported electrode with Cu*_x_*O/Ag_2_O (*x* = 1, 2) nanowires grown in-situ on a nanoporous Cu-Ag network (*C*u*_x_O*/Ag_2_O@NP-CuAg) has been successfully designed by a facile two-step approach. The integrated hierarchical porous structure, the tip-converged Cu*_x_*O/Ag_2_O nanowires combined with the interconnected porous conductive substrate, are favorable to provide more reactive sites and improve ions or electrons transportation. Compared with monometallic Cu_2_O nanowires integrated with nanoporous Cu matrix (Cu_2_O@NP-Cu), the bimetallic Cu*_x_*O/Ag_2_O@NP-CuAg composites exhibit the enhanced electrocatalytic performance for glucose. Moreover, the higher sensitivity of ~1.49 mA mM^−1^ cm^−2^ in conjunction with a wider linear range of 17 mM for the Cu*_x_*O/Ag_2_O@NP-CuAg electrode anodized for 10 min are attributed to the synergistic effect of porous structure and bimetallic Cu*_x_*O/Ag_2_O nanowires. Particularly, the integrated Cu*_x_*O/Ag_2_O@NP-CuAg composites possess good flexibility, which has been reported for the first time. Accordingly, the Cu*_x_*O/Ag_2_O@NP-CuAg with excellent glucose electrocatalytic performance and good flexibility is promising to further develop as a candidate electrode material of glucose sensors.

## 1. Introduction

In recent years, various nanostructured metals or metallic oxides have been developed as an electrode material for glucose sensors [1,2,3,4,5,6]. Among these, bimetallic Cu-Ag oxides have aroused extensive attention due to the unique synergistic effect of structure and composition [7,8]. As reported, Cu-Ag_2_O nanowalls and Cu-Ag nanocomposites displayed the improved glucose sensing performance as compared to single metal material [9,10]. However, most of them are powders and easy to agglomerate or shed from the supporting substrate, which in turn result in the inferior electrocatalytic activity and stability. It has been demonstrated that the development of a self-standing electrode is an efficient way to avoid above problems. Based on this strategy, CuO-Ag_2_O nanoparticle [11] electrode based on bulk substrate has been synthesized. It is worth noting that the 3D porous matrix is more beneficial to improve the electrocatalytic performance of materials as compared to the bulk substrate because of larger specific surface area and better connectivity. In addition, the pore network structure can obviously influence the distribution of the active substance, which further enhances the performance of material [12,13]. Regrettably, there have been rare related reports so far for the development of bimetallic Cu-Ag oxides based on 3D porous substrate. Therefore, it is of great importance to design a self-standing Cu-Ag bimetal oxides electrode with hierarchical porous structure for glucose sensing.

Herein, we have successfully synthesized the flexible free-standing Cu*_x_*O/Ag_2_O (*x* = 1, 2) nanowires electrode composites with hierarchical porous structure. The tip-converged Cu*_x_*O/Ag_2_O nanowires in-situ grown on nanoporous Cu-Ag network (NP-CuAg) were prepared via anodizing followed by calcination. Subsequently, the glucose electrocatalytic performance of as-prepared Cu*_x_*O/Ag_2_O@NP-CuAg composites was examined.

## 2. Materials and Methods

### 2.1. Synthesis of Cu_x_O/Ag_2_O@NP-CuAg Nanowire Composites

The schematic synthesis process of Cu*_x_*O/Ag_2_O@NP-CuAg (*x* = 1, 2) is depicted in Figure 1a. The Cu_42.5_Zr_50_Ag_7.5_ (at.%) metallic glass (MG) ribbon as a dealloying precursor alloy was fabricated by arc-melting and single-roller melt-spinning process [14]. Then, nanoporous copper silver (NP-CuAg) substrate was obtained by dealloying Cu_42.5_Zr_50_Ag_7.5_ ribbon in 0.05 M HF for 8 h. During the dealloying process, Zr element exhibited highly active electrochemical property in HF and selectively dissolved into HF solution [15,16,17], leaving the inert metals of Cu and Ag behind to further assemble to nanoporous bimetallic CuAg (NP-CuAg) network substrate. Finally, the anodizing measurements (Huatai, DC Power Supply, HAP 10-200, Yangzhou, China) were applied for NP-CuAg in 0.5 M KOH with the current density of 15 mA cm^−2^ for different time. The NP-CuAg and commercial Pt net electrode were employed as working and counter electrodes, respectively. Then, the anodized samples were rinsed three times with distilled water and further calcined in muffle furnace at 473 K for 2 h in the air. All experiments except for calcination were performed at 298 K. Note that the anodized samples for anodizing time of 1, 5 and 10 min are labelled as Cu*_x_*O/Ag_2_O-1, Cu*_x_*O/Ag_2_O-5 and Cu*_x_*O/Ag_2_O-10, respectively.

### 2.2. Microstructure Characterization

The phases and crystal structures of the as-prepared samples were detected by X-ray diffractometer (XRD, D8, Cu-Kα, Bruker, Karlsruhe, Germany) with the 2θ range of 25–85° and X-ray photoelectron spectroscopy (XPS, Thermo Fisher Scientific, Waltham, MA, USA). Transmission electron microscopy (TEM, JEOL JEM 2100F, Tokyo, Japan) and scanning electron microscopy (SEM, Nova nanoSEM 450, FEI, Hillsboro, OR, USA) were applied to characterize the microstructure and morphology.

### 2.3. The Electrochemical Measurements for Glucose

The electrochemical measurements were performed by the electrochemical workstation (Chenhua CHI660E, Shanghai, China) with common three-electrode system. The as-prepared samples, commercial Pt net electrode and Ag/AgCl standard electrode (3 M KCl) were employed as the work electrode, the auxiliary electrode and the reference electrode, respectively. The cyclic voltammetry measurements (CVs) and the amperometric measurements (i-t curve) were used to examine the electrochemical performance. The frequency of electrochemical impedance spectroscopy (EIS) detections was conducted in 0.2 M NaOH with 1 mM glucose ranging from 0.01 to 10^6^ Hz.

## 3. Results and Discussion

### 3.1. Design of Hierarchical Porous Structure of the Cu_x_O/Ag_2_O@NP-CuAg Electrodes

Figure 1b–d show the morphologies of the samples obtained by anodizing the NP-CuAg network substrate for different time followed by calcination. From the inset of Figure 1b, it can be observed that the as-dealloyed NP-CuAg substrate displays an open, bicontinuous ligament-channel structure with the average ligament size of ~100 nm, which can provide bimetal sources for anodizing [17]. After anodizing for 1 min (Figure 1b), the ultrafine nanowires sparsely grow on the surface of porous NP-CuAg matrix, displaying the top-converged structure. As the anodizing time prolongs to 5 min (Figure 1c), the number of nanowires grown on porous substrate increases significantly. Moreover, as seen from the magnification image in the Figure 1c, it is clearly observed that the nanowires exhibit an interesting top-converged structure feature, which coincides with the schematic of the Figure 1a. The unique structure is perhaps caused by the bending of nanowires with large aspect ratios [18] and Van der Waals’ force of the tips of nanowires. As the anodizing time further increases to 10 min (Figure 1d), the intensive nanowires grow on NP-CuAg, which offers richer active sites. The inset cross-sectional image of the sample in Figure 1d shows that nanowire layer with the thickness up to micron scale tightly combines with the porous Cu-Ag substrate, which shows a well-integration of the nanowires@porous network composites. In addition, the average length of nanowires for the Cu*_x_*O/Ag_2_O-1, Cu*_x_*O/Ag_2_O-5, Cu*_x_*O/Ag_2_O-10 is ~1.07, 1.38 and 1.73 μm, respectively. The corresponding aspect ratio of the nanowires is about ~26.75, 19.71 and 18.21, respectively. It indicates that the length of the nanowires increases with an increase in anodizing time, whereas the corresponding aspect ratio decreases.

The XRD patterns of the as-spun, as-dealloyed and anodized samples were shown in Figure 2. It is found that as-spun ribbon has the broad diffraction halo peak, showing the formation of amorphous structure. Moreover, no Zr element is detected in the as-dealloyed sample, indicating that Zr element is selectively dissolved [15,16,17], leaving the bimetals of Cu and Ag behind to assemble to nanoporous bimetallic CuAg network (NP-CuAg) (the inset of Figure 1b). After anodizing for 5 min followed by calcination, the crystalline phases of anodized sample are identified to be Cu (JCPDS #89-2838), Ag (JCPDS #04-0783), Cu_2_O (JCPDS #05-0667), CuO (JCPDS #45-0937) and Ag_2_O (JCPDS #72-2108), confirming the formation of Cu*_x_*O/Ag_2_O (*x* = 1, 2) bimetallic oxides on the NP-CuAg network.

XPS spectra (Figure 3) were further measured to determine the multi-valences of samples anodized for 5 min followed by calcination. From the typical wide-scan XPS spectrum shown in Figure 3a, it appears the presence of Cu, Ag and O in the anodized Cu*_x_*O/Ag_2_O-5 sample. The two peaks of Cu 2p_3/2_ (Figure 3b) are located at 933.1 eV and 934.57 eV together with the feature of two satellite peaks, which indicate the presence of Cu^+^ and Cu^2+^ [19]. The relative content of Cu_2_O and CuO is 41.28% and 58.72%, respectively. The Ag^+^ 3d_5/2_ and Ag^+^ 3d_3/2_ peaks (Figure 3c) located at 367.8 and 373.8 eV, respectively, with the binding energy interval of 6.0 eV are assigned to the characteristic Ag^+^ [20]. The XPS results reveal that bimetallic Cu-Ag oxide nanowires consist of CuO, Cu_2_O and Ag_2_O, which is in agreement with XRD analysis.

To obtain more detailed structural characteristics, the nanowires of the anodized sample were analyzed by TEM in Figure 4. It is observed that there are a large amount of small nanopores distributed on the nanowires with the sizes of 1~3 nm (Figure 4a). In Figure 4b, the lattice spacings of 0.246 nm, 0.253 nm and 0.234 nm, together with the corresponding selected-area electron diffraction (SAED) patterns (the inset of Figure 4b), are ascribed to the (111) plane of Cu_2_O, (002) plane of CuO and (011) plane of Ag_2_O, respectively, indicating that the nanowires are composed of CuO, Cu_2_O and Ag_2_O. The mapping images (Figure 4c–e) further reveal that O, Cu and Ag atoms uniformly distribute on the nanowires. Based on the above results, it can be concluded that the bimetallic nanowires that consist mainly of Cu*_x_*O mixed with minor Ag_2_O, integrated with porous Cu-Ag network have been successfully prepared by a facile two-step method. Furthermore, the unique features of the multivalent states (Cu^+^/Cu^2+^ and Ag^+^), the hierarchical porous structure as well as the integrate design without any additional binders endow with the enhanced glucose electrocatalytic performance of Cu*_x_*O/Ag_2_O@NP-CuAg composite.

### 3.2. Electrocatalytic Performance of the Cu_x_O/Ag_2_O@NP-CuAg Electrode Composites

As compared to the free-standing monometallic Cu_2_O nanowires composite (Cu_2_O@NP-Cu) [19], it should be emphasized that the new Cu*_x_*O/Ag_2_O@NP-CuAg electrodes not only keep good mechanical integrity, but also could withstand a large degree of bending (inset photo of Figure 5a), indicating a good flexibility. The CVs measurements in 0.2 M NaOH with 3 mM glucose (Figure 5a) are performed for the flexible Cu*_x_*O/Ag_2_O@NP-CuAg electrode before and after bending. It appears that both the CVs almost coincide to each other. Moreover, the oxidation peak appeared at 0.5 V is clearly higher as compared to that in the NaOH without glucose, which demonstrates the electrode modified by bimetallic Cu*_x_*O/Ag_2_O oxide nanowires possesses high glucose sensing performance.

Amperemetric i-t curves (Figure 5b) of the bimetallic Cu*_x_*O/Ag_2_O@NP-CuAg electrodes anodized for different time were conducted in the 0.2 M NaOH solutions by dropping 1 mM glucose at 0.5 V, while that of monometallic Cu_2_O@NP-Cu electrode anodized for 5 min [19] is also shown for comparison. From the i-t curves, it can clearly see that all of the electrodes have significant current response signals when 1 mM glucose is added. The corresponding linear fitting of the current density plotted with the addition of glucose concentration was shown in Figure 5c. It is found that all the Cu*_x_*O/Ag_2_O@NP-CuAg electrodes exhibit wide linear range up to 17 mM. Additionally, the sensitivity value fitted by software for the Cu*_x_*O/Ag_2_O-10, Cu*_x_*O/Ag_2_O-5, Cu*_x_*O/Ag_2_O-1, and Cu_2_O composite electrodes is ~1.49, 1.38, 0.85, and 0.58 mA mM^−1^ cm^−2^, respectively. The corresponding standard error of them is ~0.028, 0.026, 0.020, and 0.012, respectively. Thus, the Cu*_x_*O/Ag_2_O@NP-CuAg electrode after anodizing 10 min has the highest sensitivity value. Moreover, as compared to monometallic Cu_2_O@NP-Cu electrode, it is found that the bimetallic Cu*_x_*O/Ag_2_O@NP-CuAg electrodes possess higher sensitivity and wider linear range. The Nyquist plots (Figure 5d) display that the radius of the semicircle for electrode increases with an increase in the anodizing time, indicating that the extension of anodizing time results in an enhancement in the electron transfer resistance of the electrode [21]. However, it is seen from Figure 5c that the Cu*_x_*O/Ag_2_O-10 exhibits the highest sensitivity. Thus, the effect of the increased electron transfer resistance is far less than the improvement of electrocatalytic performance caused by synergistic effect of hierarchical structure and bimetallic oxides.

Generally, glucose often coexists with small amounts of interfering substances such as ascorbic acid (AA) and uric acid (UA) in human blood. In order to investigate the anti-interference ability of the present electrode, the i-t curve of Cu*_x_*O/Ag_2_O-10 was measured by adding 0.1 mM AA, 0.1 mM UA and 3 mM glucose in 0.2 M NaOH solution at 0.5 V, respectively [22]. As shown in Figure 6a, the addition of glucose results in a significant current response for Cu*_x_*O/Ag_2_O-10, whereas the current signals of AA and UA are neglectable, illustrating that the present composites demonstrate the excellent anti-interference ability towards glucose detection in human blood. In addition, the stability of the electrode is also important for its practical application. Herein, the Cu*_x_*O/Ag_2_O-10 electrode was exposed in the air, and the current response to 1 mM glucose was monitored once a day. Note that the current intensity measured daily is normalized with the initial value and the result is shown in the Figure 6b. It is found that the current response detected for 30 days still remains about 97.01%, revealing that the new electrodes demonstrate outstanding long-term stability, which may derive from its integrated structural advantages of in situ growth.

## 4. Conclusions

In this work, the free-standing Cu*_x_*O/Ag_2_O@NP-CuAg electrode with good flexibility has been fabricated by a facile fabrication strategy. The bimetallic Cu*_x_*O/Ag_2_O nanowires integrated with the nanoporous Cu-Ag network present unique structure features, i.e., the tip-converged nanowires and the hierarchical porous structure. SEM results show that the length of the nanowires increases with an increase in anodizing time, whereas the corresponding aspect ratio decreases. In addition, as compared to monometallic Cu_2_O@NP-Cu electrode, it is found that the electrodes modified by bimetallic Cu*_x_*O/Ag_2_O oxide nanowires demonstrate much higher glucose sensing performance. The Cu*_x_*O/Ag_2_O@NP-CuAg electrode anodized for 10 min demonstrates the highest sensitivity of ~1.49 mA mM^−1^ cm^−2^, wide linear range up to 17 mM as well as outstanding long-term stability. The high electrocatalytic performance of glucose for the new flexible electrode is attributed to the integrated hierarchical porous structure and the synergistic effect of the copper and silver elements. The newly developed Cu*_x_*O/Ag_2_O@NP-CuAg composite is a prospective candidate for flexible glucose sensor.

## Figures and Tables

**Figure 1 nanomaterials-10-00357-f001:**
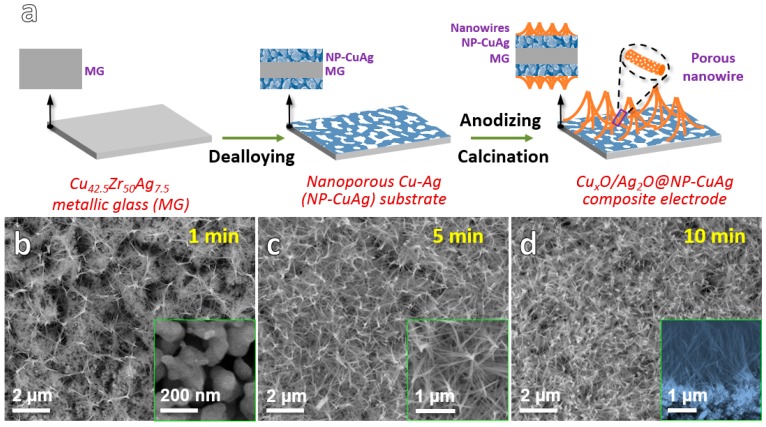
(**a**) Schematic illustration showing the synthesis process of Cu*_x_*O/Ag_2_O@NP-CuAg composite; SEM images of Cu*_x_*O/Ag_2_O@NP-CuAg anodized for different time followed by calcination (**b**) 1 min with the inset of NP-CuAg substrate; (**c**) 5 min and the magnification image; (**d**) 10 min with the sectional image.

**Figure 2 nanomaterials-10-00357-f002:**
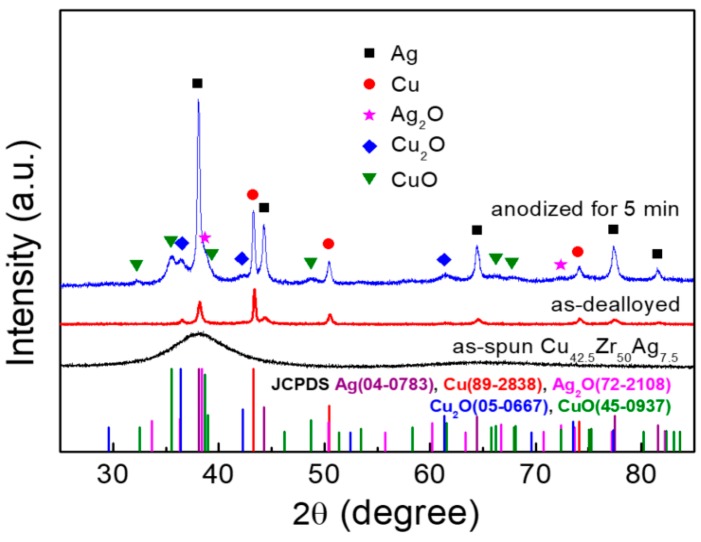
XRD patterns of the as-spun, as-dealloyed and as-prepared sample after anodizing for 5 min followed by calcination.

**Figure 3 nanomaterials-10-00357-f003:**
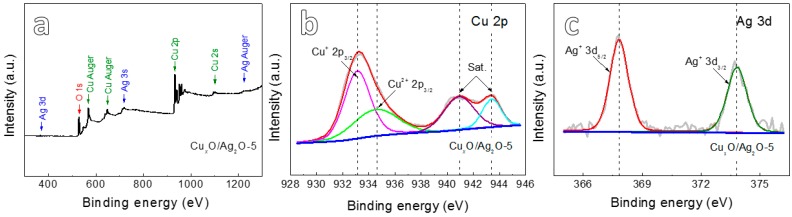
(**a**) XPS spectrum of the Cu*_x_*O/Ag_2_O-5; (**b**) Cu 2p XPS spectrum of Cu*_x_*O/Ag_2_O-5; (**c**) Ag 3d XPS spectrum of Cu*_x_*O/Ag_2_O-5.

**Figure 4 nanomaterials-10-00357-f004:**
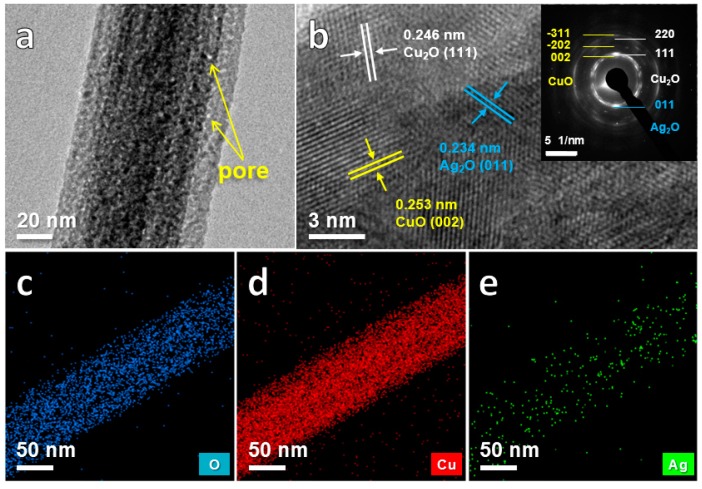
TEM images of the Cu*_x_*O/Ag_2_O-5 nanowires (**a**) high magnification image; (**b**) HRTEM image with the SAED pattern; the EDS elemental mapping images of (**c**) O; (**d**) Cu; (**e**) Ag.

**Figure 5 nanomaterials-10-00357-f005:**
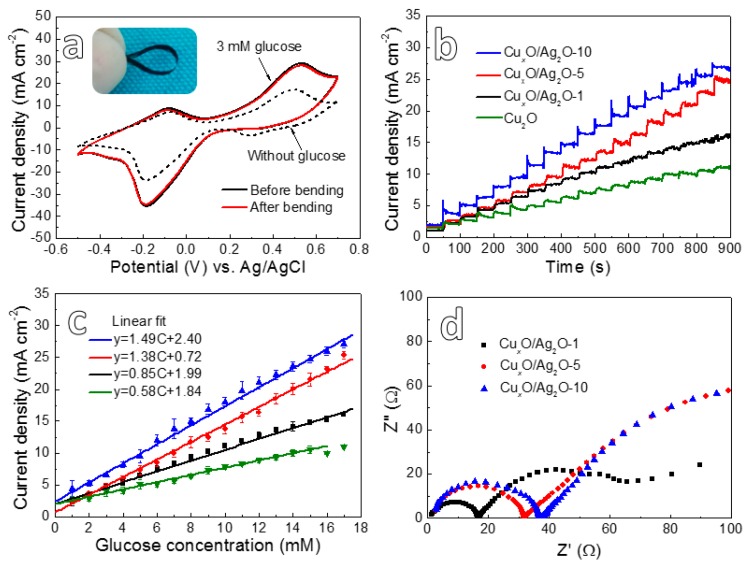
(**a**) The CVs of Cu*_x_*O/Ag_2_O-10 in 0.2 M NaOH solution with or without 3 mM glucose over the potential range from -0.5 V and 0.7 V at a scan rate of 20 mV s^−1^ with the photo of electrode after bending; (**b**) amperometric i-t curve of Cu*_x_*O/Ag_2_O@NP-CuAg electrodes with successive additions of 1 mM glucose into 0.2 M NaOH solution at an applied potential of 0.5 V; (**c**) the corresponding glucose calibration curves for (**b**); (**d**) EIS of the samples anodized for different time in 0.2 M NaOH with 1 mM glucose.

**Figure 6 nanomaterials-10-00357-f006:**
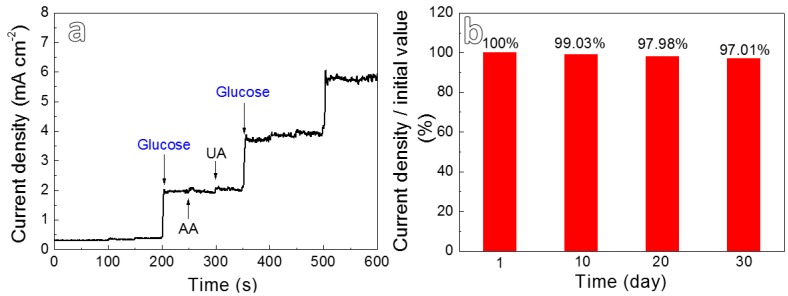
(**a**) Anti-interference ability measurement of Cu*_x_*O/Ag_2_O-10 at an applied potential of 0.5 V; (**b**) the stability of the Cu*_x_*O/Ag_2_O-10 electrode over a period of 30 days.

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
