# Peer review of "Flexible Free-Standing CuxO/Ag2O (x = 1, 2) Nanowires Integrated with Nanoporous Cu-Ag Network Composite for Glucose Sensing"

_nanomaterials, 2020, doi:10.3390/nano10020357_

Round 1
Reviewer 1 Report
The Authors report on a glucose sensor based on a hierarchical structure of free-standing (tip converged) copper oxide/ silver oxide nanowires grown by anodization process with three treatment time conditions, (optimum time 10min) on the underlying submicron Cu/Ag particles. The sensing performance of the nanomaterial is compared with the similar silver-free catalyst (CuOx@Cu) and an improvement is clear both on the sensitivity, durability and flexibility.
The paper is interesting and characterizations are complete and convincing, but needs some additional work to improve its impact:
The text lacks of accuracy and should be edited from a Native English scientist; just few examples from the first page:
In the abstract, "the electrocatalytic performance of glucose". This is a non-sense, to my understanding, the meaning would be something like "the sensing activity of the catalyst material versus the glucose detection".
Few lines later, "fertile active sites" active for what?
Later in the introduction "Cu-Ag nanocomposites display higher catalysis", what is a higher catalysis? Is at a catalytic activity, versus which reaction?
Line 34 "bring about inferior"; lowers?
Etc.
Figure 1 has to be revised. Please precise what is MG. The starting submicron particles need to be characterized more in details (average size and morphology). Where goes the Zr? It is surprising not to see Zr traces in XPS survey spectra.
What is the average length or aspect ratio of the wires vs. anodization time?
The XPS analysis lacks of quantification of elements. Also, the Cu2p HR spectrum background looks much too steep. I suggest to show more clearly the experimental data and the fit peaks sum.
Line 149, when comparing the glucose detection sensitivities, the 10 min sample is described as the one showing the "highest sensitivity of 1.5mA", what is the difference between samples, a quantification has to be provided on electrochemical data. Provide détails on errors on these data.
Figure 6b I do not understand the choice of the red bars, I would keep the points and zoom the y-scale.
Reviewer 2 Report
this manuscript reports a careful study about a novel composite electrode material for glucose sensing. the authors illustrate in a compact and complete way this innovation where silver oxide is employed as coactive material with copper oxide.
overall the manuscript is well done and the results fully support the hypothesis. I have one specific point that requires attention and clarification:
in the section 3.1 it is not clear if the xrd sem and xps refer to annealed samples or to films obtained after amortization.please clarify.
once clarified this manuscript can be accepted.
Round 2
Reviewer 1 Report
I can see a clear improvement of the MS accuracy. I do not see additional concerns and I suggest to publish the paper in its present form.
Regards